# Prioritizing autoimmunity risk variants for functional analyses by fine-mapping mutations under natural selection

Vasili Pankratov [1,5], Milyausha Yunusbaeva [2,5], Sergei Ryakhovsky [2], Maksym Zarodniuk [3], Estonian Biobank Research Team* & Bayazit Yunusbayev [1,2] ✉

Pathogen-driven selection shaped adaptive mutations in immunity genes, including those contributing to inflammatory disorders. Functional characterization of such adaptive variants can shed light on disease biology and past adaptations. This popular idea, however, was difficult to test due to challenges in pinpointing adaptive mutations in selection footprints. In this study, using a local-tree-based approach, we show that 28% of risk loci (153/535) in 21 inflammatory disorders bear footprints of moderate and weak selection, and part of them are population specific. Weak selection footprints allow partial fine-mapping, and we show that in 19% (29/153) of the risk loci under selection, candidate disease variants are hitchhikers, and only in 39% of cases they are likely selection targets. We predict function for a subset of these selected SNPs and highlight examples of antagonistic pleiotropy. We conclude by offering disease variants under selection that can be tested functionally using infectious agents and other stressors to decipher the poorly understood link between environmental stressors and genetic risk in inflammatory conditions.

Pathogens exerted strong selective pressure on human immune traits[1,2]. Detecting the genetic footprints of these selection events can help us identify genotypes that were important for survival earlier in life and understand their later-life adverse consequences for immune-related diseases. This is a half-century-old idea of antagonistic pleiotropy. It was initially proposed to explain ageing[3,4] but later was invoked to explain autoimmunity[5,6] and other human traits[7]. Indeed, genomic evidence suggests that immune-related genes were targets of natural selection[8–12]. Accordingly, genetic risk loci for autoimmune diseases also bear signals of natural selection, represented by extended haplotypes[12,13]. These findings, however, can be explained by two competing models: causal variants in these autoimmunity risk loci were driving selection signals, or they were in linkage disequilibrium with mutations undergoing selective sweep, i.e., hitchhiking[5,14]. If causal variants were driving selection

signals, they are expected to have a tangible effect on underlying molecular traits and be easier to detect and study experimentally. Such variants are promising for functional analyses to understand their role in the pathogenesis of immune-mediated diseases[13–17].

Given that some risk SNPs have adaptive history, an important next step is to understand their function by testing various environmental factors, assuming a connection between the selective agent and disease trigger. For example, microbial exposure exerted strong selective pressure on immunity genes both in humans[1,18,19] and other organisms[20,21] and at the same time is known to trigger many autoimmune diseases[22,23]. Hence, it can be a useful key for discovering risk allele function in early pathogenic events.

Currently, there are only a few examples suggesting a potential link between natural selection, resistance to pathogens and disease

[1]University of Tartu, Institute of Genomics, Centre for Genomics, Evolution and Medicine, Tartu 51010, Estonia. [2]ITMO University, SCAMT Institute, Saint-Petersburg 191002, Russia. [3]University of Tartu, Institute of Bio- and Translational Medicine, Tartu 50411, Estonia. [5]These authors contributed equally: Vasili Pankratov, Milyausha Yunusbaeva. *A list of authors and their affiliations appears at the end of the paper. ✉e-mail: yunusbb@gmail.com

risk. For example, celiac disease risk SNP rs3184504 in the *SH2B3* gene and Crohn's disease SNP rs601338 in the *FUT2* gene, where we have evidence that candidate causal variants were under selection and modulate resistance to pathogens[24–27]. This hypothetical link between natural selection and disease risk remains unexplored since fine-mapping the adaptive mutation in the selective sweep signal is challenging. Earlier works that tested autoimmune risk loci for selection used ad-hoc significance thresholds, chip genotyping data with low SNP resolution (one SNP every 6–10 kb, on average)[12,13], and lacked powerful methods to localize adaptive mutations, so-called selection targets[28]. That, however, has changed recently with key methodological improvements[29,30]. More importantly, earlier works that focused on autoimmune risk loci[12,13] could detect only strong signals of selective sweeps, which are rare events and leave little genetic variation to fine-map the sweep-driving mutations, the selection targets. In contrast, weak and moderate selective sweeps are harder to detect, but once identified, their sweep-driving SNP is easier to locate. Therefore, an important question is whether selection targets related to immune phenotypes and inflammatory conditions evolved under strong, moderate, or weak selection regimens and can be pinpointed.

In this study, we report key progress in understanding the evolutionary history of the genetic component of inflammatory conditions. We use a recently proposed Relate approach[31] to build local trees for candidate risk SNPs and then the CLUES method to evaluate evidence for natural selection by computing the relative likelihood of selection scenario over neutrality, coefficient of selection, and allele frequency trajectory[32]. This powerful approach was applied to a large dataset of over 2300 fully sequenced genomes of Estonian Biobank[33,34]. We show that whenever risk loci associated with inflammatory conditions show evidence for natural selection, selective sweep intensity was likely weak or moderate. This is an important finding since it facilitates fine-mapping the selection target in the sweeping region. We fine-map selection targets in 153 risk loci that show evidence for selection. This allowed us to distinguish evolutionary scenarios underlying selection sweep signals within analyzed risk loci, unlike previous haplotype-based approaches. Namely, evolutionary scenarios, where candidate causal SNPs likely evolved under positive selection or were hitchhiking, that is, in linkage disequilibrium with adaptive mutation. Thus, the key contribution is that we identify candidate causal SNPs under selection that are promising candidates for experimental study. In addition, we suggest that microbial exposure can be relevant context to model SNP function since microbial agents are well known to trigger autoimmunity[22,23] and were likely selective agents among other stressors. This is an important step to overcoming the current challenges of finding the disease variants and relevant physiological context to study their function in pathogenesis.

## Results

### Evidence for natural selection corrected for population demography

To test whether autoimmunity-associated loci evolved under natural selection, we started with 4331 SNPs that were reported by ref. 35 as candidate causal SNPs in 535 unique GWAS hits ("index" SNPs) for 21 autoimmune conditions (Supplementary Fig. 1). Each candidate SNP in Farh et al.[35] was reported with PICS score (Probabilistic Identification of Causal SNPs) reflecting its probability to be causal. We included additional 6156 SNPs from Estonian Biobank whole-genome sequences if they were in high LD (linkage disequilibrium measured as $r2 \geq 0.8$) with the 4331 SNPs from ref. 35. This resulted in 10487 SNPs that we refer to as candidate causal SNPs throughout the text (Supplementary Data 1). For natural selection test, we used the CLUES tool version 1.0[32], which assesses the goodness-of-fit between natural selection and neutrality scenarios for the tested SNP, i.e., perform likelihood ratio test (logLR). Specifically, CLUES can use the properties of the SNP genealogy (local tree) to infer the allele frequency trajectory:

increasing and decreasing trajectories yield positive or negative selection scenarios as well as the selection coefficient (see Methods for the exact interpretation of s).

To minimize spurious selection signals due to genetic drift in the history of the studied population, we simulated the demographic history of the Estonian population and estimated the logLR cutoff (Supplementary Fig. 2a, b). Namely, we used the 95% percentile of the simulation-based logLR distribution (logLR ≥ 1.59) as our neutrality rejection threshold (Supplementary Fig. 2b).

We tested evidence for natural selection for a subset of 9102 candidate SNPs (out of the 10487 candidate causal SNPs in high LD) if they mapped on Relate-inferred local trees and passed all the filtering criteria for CLUES. These tested SNPs represent candidate causal variants for 535 risk loci that fall into 464 LD blocks ($r2 \geq 0.6$) (Supplementary Data 2). On average, each risk locus was represented by 17 tested SNPs (minimum one and maximum 158). If tested SNPs had a median logLR above the cutoff of 1.59, we additionally required nearby SNPs to have consistently high values to suggest a selection signal. In this way, we identified 94 LD blocks containing 153 risk loci with SNPs showing consistent selection signal (logLR ≥ 1.59). We, therefore, infer that ~28% (153/535) of the risk loci for various inflammatory conditions contain at least one or two SNPs that demonstrate consistent evidence for natural selection.

### Strength of selection and mappability of mutations driving selection

Although traces of strong (high values of s) selective sweeps are easier to detect at the genomic level, fine-mapping sweep-driving mutations within such traces, i.e., sweep regions, is harder. Strong selection quickly brings the selected haplotype with linked SNPs to high frequency, thereby removing local SNP variation and leaving less time for recombination to restore flanking SNP variation that would be informative for fine-mapping. In contrast, weaker selective sweeps allow for more recombination and create local differences in SNP variation, making fine-mapping feasible. We, therefore, explored the distribution of selection coefficients (summarizes strength of selection over a time period) among tested candidate SNPs (Fig. 1a) to learn the proportion of risk loci where fine-mapping is feasible. When we focused on the SNPs in the 153 risk loci that have consistent evidence for selection, we found that tested SNPs mostly corresponded to sweep signals with moderate selection coefficients (circles and bars in orange, Fig. 1a, b), and only a few SNPs demonstrated strong selection sweep. Here, we define strong, moderate, and weak selection events following the Schrider and Kern work[36]. These findings suggest that our candidate SNPs for downstream analyses changed in frequency relatively slowly, and there was time for recombination to counteract the sweeping effect. Our estimates, therefore, suggest that the local patterns of logLR variation within the reported 153 risk loci can be informative in prioritizing sweep-driving SNPs.

Fine-mapping selected SNP using local logLR variation can attain very high resolution. This is because within a given haplotype block under selection, local trees can be inferred for very small chunks bounded by recombination. Consecutive local trees, while correlated in topology and branch length, would still slightly differ from one another due to historical recombination(s). Therefore, sweep-driving SNPs separated from linked neutral alleles by recombination are expected to yield higher logLR values. This principle was used in the original study[32] to fine-map the selection targets within sweep regions.

Next, we also inferred the approximate age of each tested SNP. We show that most of the candidate SNPs, including a subset with evidence for natural selection (significant at logLR≥1.59), arose before the out-of-Africa event (Fig. 1c). As expected, strongly selected SNPs (on the bottom right) all have relatively recent allele ages since older ones quickly reached fixation and are absent in the analyzed polymorphic SNP set.

## Population-specific selection in disease risk loci

Since most of the candidate SNPs and selection targets in the Estonian population arose before out-of-Africa, we expect they are partly shared with other European populations. For comparison, we considered genetically close Finns (FIN) and more distant British (GBR) and Italians (TSI). Figure 1d shows that most candidate SNPs and top disease candidates with PICS ≥ 0.5 have similar allele frequencies in European populations. Next, we tested all the candidate SNPs for selection in Finns, British and Italians using Relate/CLUES approach and compared concordance (Fig. 1e). For most candidate SNPs that

likely evolved neutrally in Estonians (data points in black at the lower left corner, Fig. 1e), logLR values in other populations were small and varied randomly, and hence, we observed a low correlation. We then focused on a subset of SNPs that showed increasing evidence for selection in Estonians (at logLR ≥ 1.59 and logLR ≥ 3) to see if selection signals are shared with other European populations. While there was an increase in correlation with increasing evidence for selection at logLR ≥ 1.59 (data points in orange, Fig. 1e) and logLR ≥ 3.0 (Supplementary Fig. 3), the overall agreement between populations was low. Low reproducibility across populations might be due to the

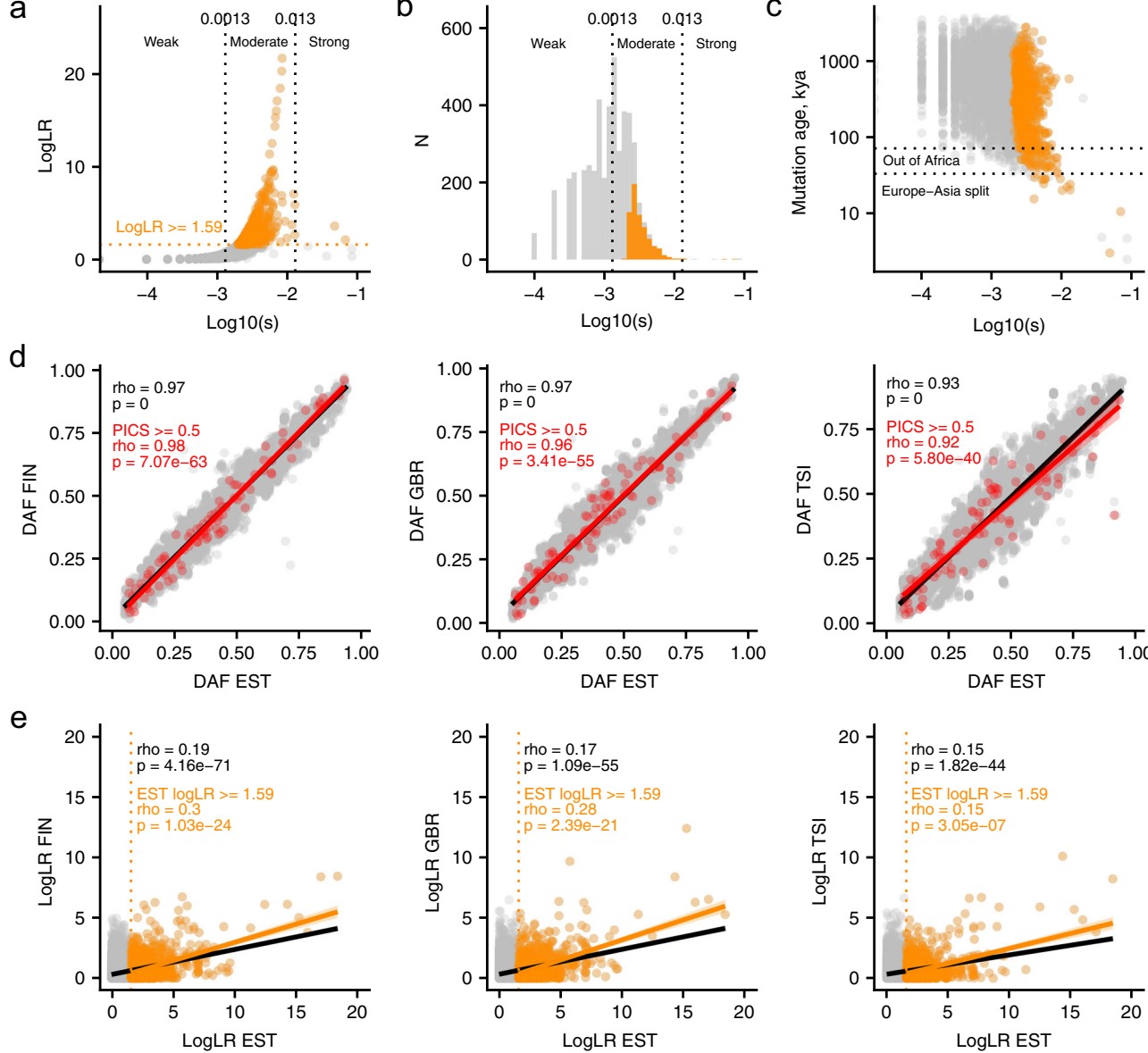

**Fig. 1 | Relationship between logLR, selection coefficient, and allele age for 9102 candidate SNPs and reproducibility of selection signals between populations. a** Evidence for selection (logLR, Log-transformed likelihood ratio of the selection scenario) versus the selection coefficient. The horizontal dashed line in orange indicates the neutrality rejection threshold. Vertical dashed lines (in gray) separate weak, moderate and strong selection coefficients, following the Schrider and Kern work[36]. **b** SNP counts with weak, moderate and strong selection coefficients among neutral and significant calls (above neutrality threshold, logLR ≥ 1.59). **c** Derived allele age for tested candidate SNPs and their selection coefficients. SNPs with evidence for selection (logLR ≥ 1.59) are shown in orange. **d** Spearman correlation (rho) of candidate SNPs' derived allele frequencies between populations; frequencies in Estonians are compared with Finns (FIN), British (GBR) and Italians

(TSI) from the 1000 Genomes Project. Rho and p-values (upper left corner) for the total SNP set are in black, and for most likely causal SNPs (PICS > 0.5) are in red. **e** SNP-wise logLR estimates in Estonians are compared to that in Finns (FIN), British (GBR) and Italians (TSI). Spearman correlation (rho and p-value in orange) is computed separately for SNPs (1105 out of 9102) with evidence for selection in Estonians (points in range, logLR ≥ 1.59) and the total set of 9102 SNPs (rho and p-value in black). Nominal p-values are reported with no multiple testing correction. In both **d** and **e** the trend line is obtained by fitting a generalized additive model to the corresponding data and the shaded area corresponds to its' 95% confidence interval. Raw data for all the figure panels are provided in Supplementary Data 2.

population-specificity of weak to moderate selection signals and variance in logLR estimates.

## Prioritizing selection targets among candidate causal variants

It was shown in the original study that the logLR could be used to fine map the sweep-driving SNP[32], and the mapping accuracy is only limited by the extent of linkage disequilibrium (See Fig. 8A in ref. [32]). We examined the logLR values for a consecutive set of candidate SNPs within the 153 risk loci and picked the highest scores to pinpoint the SNP(s) that likely drove the selection signal. We then asked whether these predicted selection target SNPs matched the likely causal variants within the risk locus. For that, the highest logLR values were compared with the highest PICS (Probabilistic Identification of Causal SNPs) scores; the latter estimates the probability of the candidate SNP to be causal[35]. We identified a range of scenarios in this way (Fig. 2b), and full results (graphs analyzed) for the 153 risk loci are available in the Figshare repository (See Data availability statement and Supplementary Data 3). For example, the most likely selection target SNPs and their alleles matched the most likely causal variants and effect alleles (Fig. 3a) or cases where the selection target SNP allele corresponded to the protective allele of the top causal variant (Fig. 3b). These two scenarios identify SNPs that are promising targets for downstream functional analyses, and we report all such SNPs in Supplementary Data 2 ("Promising SNPs") and their LD blocks in Supplementary Data 3. However, we often found that the prioritized selection targets just happened to be nearby the top candidate variant (Fig. 3c, d). In such cases, we looked at the haplotype phase and further defined sub-scenarios (Fig. 3c, d). For example, in one case, the top candidate risk allele for Multiple Sclerosis was hitchhiking with the selected allele, and in another case (Fig. 3d), the top candidate protective allele for Vitiligo was hitchhiking with the selected allele.

Our high-resolution analysis suggests that some of the 153 risk loci contain multiple sweep signals that represent a composite of evolutionary scenarios. Since selection signals often appear as an extended region of depleted diversity, such complex scenarios could have been missed by previous haplotype-based tests. For example, a causal SNP might appear as a selection target locally, but our sequence data combined with the Relate/CLUES approach can distinguish stronger sweep signals nearby, attaining higher resolution (Supplementary Fig. 4). In this case, one cannot rule out hitchhiking. We distinguish such complex scenarios (Scenario C in Fig. 2) from clear examples of hitchhiking (Scenario B in Fig. 2) and adaptive history for candidate causal SNPs (Scenario A in Fig. 2). We additionally highlighted seven risk loci, which could not be assigned to any of the A, B, and C major scenarios since logLR estimates were missing for the top candidate SNPs (D in Fig. 2). Figure 2 summarizes all the possible scenarios, including complex ones and the number of times they were encountered (See Supplementary Data 3 an d Data availability statement for the annotation graphs for the 153 risk loci used for this summary). Thus, our high-resolution analysis shows that genomic signals of natural selection in disease risk loci stem from various evolutionary scenarios that could not be recognized in earlier studies.

## Revisiting previous studies and the role of genetic hitchhiking

It has been long hypothesized that causal alleles for inflammatory conditions were subject to genetic hitchhiking due to linkage disequilibrium with selected SNPs. With a few empirical examples, the role of this evolutionary scenario is poorly understood[14]. We analyzed the largest collection of fine-mapped candidate SNPs for the largest collection of inflammatory conditions. We found that in genomic risk loci with selection, genetic hitchhiking (29/153, Fig. 2b) is quite frequent and strong LD ($r^2 > 0.6$) often complicates the identification of selection target SNPs (57/153, see scenario C in Fig. 2b). We also revisited earlier studies that reported positive selection for disease risk alleles but which had limited resolution in terms of data used

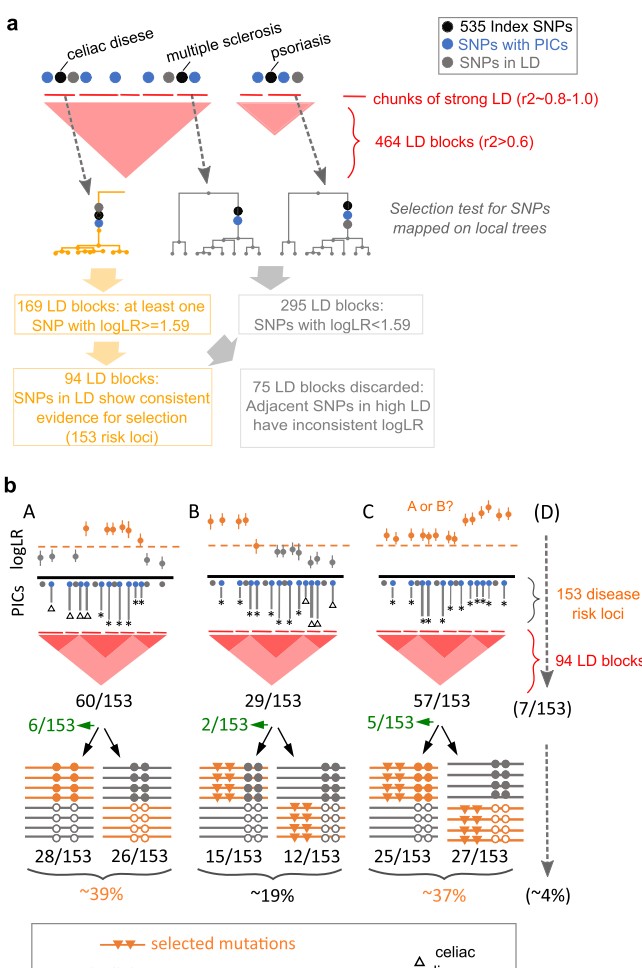

**Fig. 2 | Evolutionary scenarios for candidate causal SNPs based on logLR fine-mapping. a** Spatial organization of disease risk loci and selection test workflow. Single disease risk loci (asthma and psoriasis) and multiple disease risk loci within LD blocks (triangular heatmap) are shown schematically. Selection tests are performed for each small chunk (red segments above triangular heatmap) within disease risk loci. Nearby chunks, depending on local LD, have correlated local trees and evidence for selection. We used this fact to additionally filter out spurious selection signals that are not consistent with nearby chunks. For example, a single chunk with logLR = 2.5 stands out from adjacent chunks with logLR = 0.5. **b** Evolutionary scenarios based on comparing SNPs with top logLR and PICS. The top row schematically shows three evolutionary scenarios (A, B, C) discernible for the 153 disease risk loci with a consistent signal of selection: (A) top candidate causal variant(s) (with the highest PICS value) is/are likely sweep-driving mutation(s) (SNP(s) with the highest logLR value); (B) top candidate causal variant(s) is/are hitchhiking; (C) complex scenario: top candidate causal variant is either sweep-driving mutation itself or hitchhiking with a stronger target of selection. Black arrows further classify evolutionary scenarios at the haplotype level (shown schematically on the bottom row) by clarifying whether selection and hitchhiking affected the risk or protective allele. Arrows and numbers in green summarize cases where allele information is missing. We, therefore, distinguish six sub-categories at the haplotype level that are depicted on the bottom row. Full annotation graphs for the 153 risk loci classified in this figure are available in the Figshare repository (See Data availability statement). Raw data for this figure are provided in Supplementary Data 3.

(haplotype data) and methodology. For example, one of the pioneering works suggested that the celiac disease rs3184504*A variant in the *SH2B3* gene had adaptive history and possibly played a role in pathogen resistance[27]. Zhernakova et al.[27] showed that this SNP variant demonstrated strong evidence for positive selection based on the

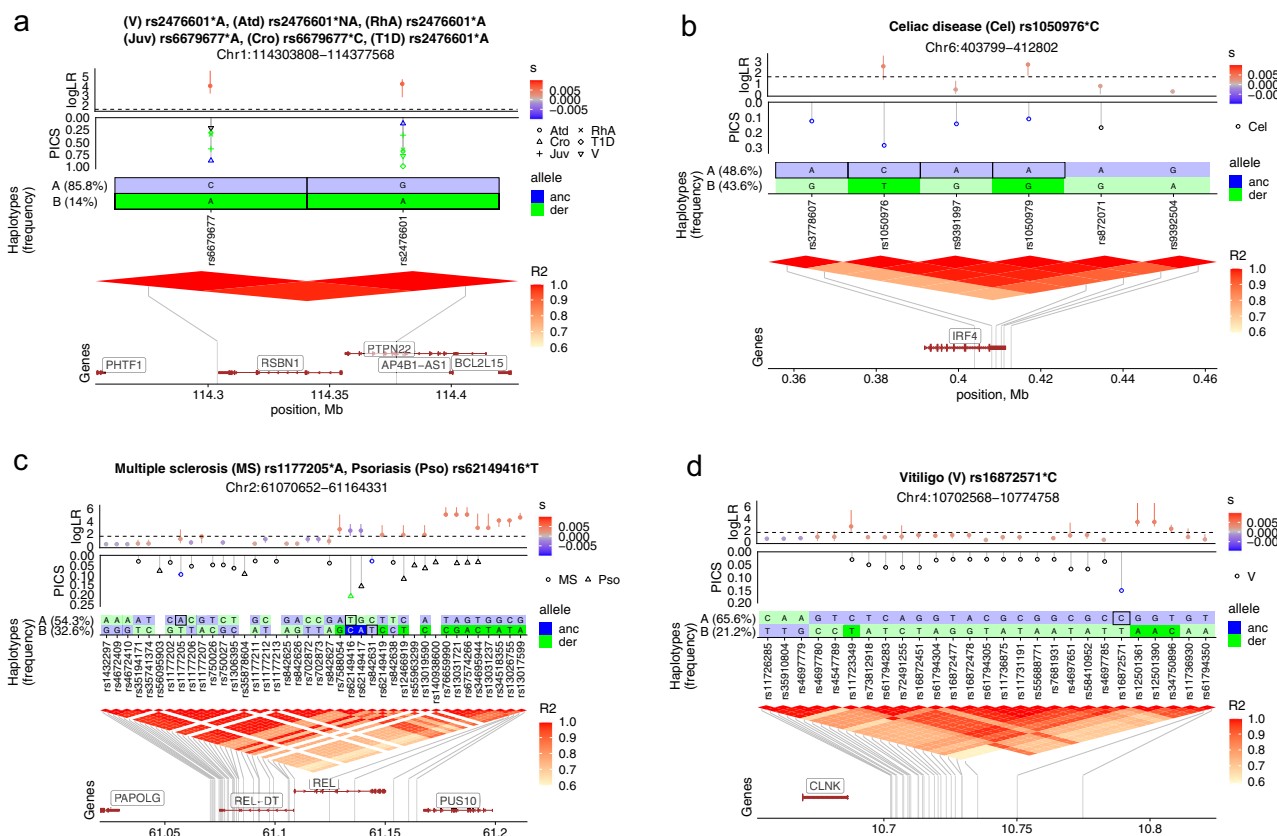

**Fig. 3 | Selection targets and hitchhikers among candidate risk SNPs.**
**a**, **b** Disease risk and protective alleles are likely selection targets. **c**, **d** Disease risk and protective alleles are hitchhikers. Graph rows represent five layers of annotation, from top to down - logLR, PICS, Haplotypes, LD heatmap, and Genes. The top row shows median logLR estimates with whiskers indicating minimum and maximum of three estimates. Whisker color reflects the sign and magnitude of the selection coefficient estimated on the SNP derived allele. The dashed line separates LogLR values (below) expected under the neutral demographic history of Estonians. The values above suggest evidence for selection. The second row shows PICS values for each candidate SNP reported in ref. 35. Candidate SNPs for multiple diseases are shown with different symbols. The symbol is blue when the risk allele is ancestral and green when derived. The next row shows the haplotype structure in the LD-block with respect to ancestral (blue) and derived alleles (green). Only the two most common haplotypes (A and B) are shown with frequencies (in brackets) estimated in 1800 Estonians. SNPs with logLR ≥ 1.59 are highlighted with darker colors, and risk alleles are indicated with solid margins. The fourth row shows a heatmap of pairwise linkage disequilibrium between all the candidate SNPs from ref. 35 even if they have r² ≤ 0.6, which defines an LD-block region. The bottom row depicts gene annotations from the Ensembl genome database (version 87, Human genome build GRCh37). Raw data for all the figure panels are provided in Supplementary Data 2.

haplotype-based iHS statistics (iHS = −2.597, *p* = 0.009). Monocytes with this risk allele were demonstrated to have stronger, dose-dependent expression of 3 inflammatory cytokines in response to bacterial ligands[27]. We reanalyzed this risk locus and confirmed that the rs3184504*A variant was under strong positive selection (rs3184504, logLR = 16.8512) (Supplementary Fig. 5). However, in our sequence data, we found novel variants in strong LD with the rs3184504 SNP (r2 ≥ 0.9–1) that demonstrated comparable (rs10774624, logLR = 15.28) and somewhat stronger evidence for positive selection (rs653178, logLR = 21.97) (Supplementary Fig. 5). Thus, with higher resolution, our analyses suggest that other SNPs in strong LD are equally possible selection targets, and further work is needed to verify the precise adaptive scenario for this genomic locus.

We next revisited one of the most comprehensive studies that suggested a natural selection history for genes implicated in inflammatory diseases[13]. Raj et al.[13] tested 416 risk SNPs for inflammatory conditions using the haplotype-based iHS-score. They reported 21 SNPs with extreme iHS-scores (|iHS| ≥ 2) and suggested a positive selection scenario for ~5% of the risk loci (21/416). We analyzed these SNP regions and found that for most of these loci, CLUES (14 out of 21 SNPs had logLR < 1.59) yield surprisingly weak support for natural selection, with logLR varying between 0.04–1.44. Only 7 out of the 21

previously reported SNPs showed evidence for natural selection at logLR ≥ 1.59, and logLR estimates suggested relatively strong selection (Supplementary Data 4). As suggested before, part of the discrepancy might be due to the population specificity of positive selection. In addition, picking extreme iHS scores (usually top 5%) does not guarantee a distinction between neutral drift and selection. Since logLR tends to focus on smaller regions bounded by recombination and the iHS score summarizes haplotype diversity over longer genomic regions, we wanted to clarify the selection signal space where these two methods would agree. We compared iHS-scores, and logLR values computed on the same Estonian Biobank sequence data. We found that iHS scores did not follow increasing logLR values over much of the logLR space (logLR between 1.59 and 10, Supplementary Fig. 6) but agreed with logLR estimates (followed the linear fitted line) only for the strongest selection signals (values over logLR > 12, Supplementary Fig. 6e). which, on average, span longer genomic regions.

The positive and negative selection that we inferred represents only two possible scenarios for the complex trait evolution. Both theory and experimental evolution suggest that genetic loci involved in resistance to pathogens might follow a frequency-dependent selection[37]. This selection regime creates the so-called balancing haplotypes. The best-known balancing haplotypes in humans (genomic

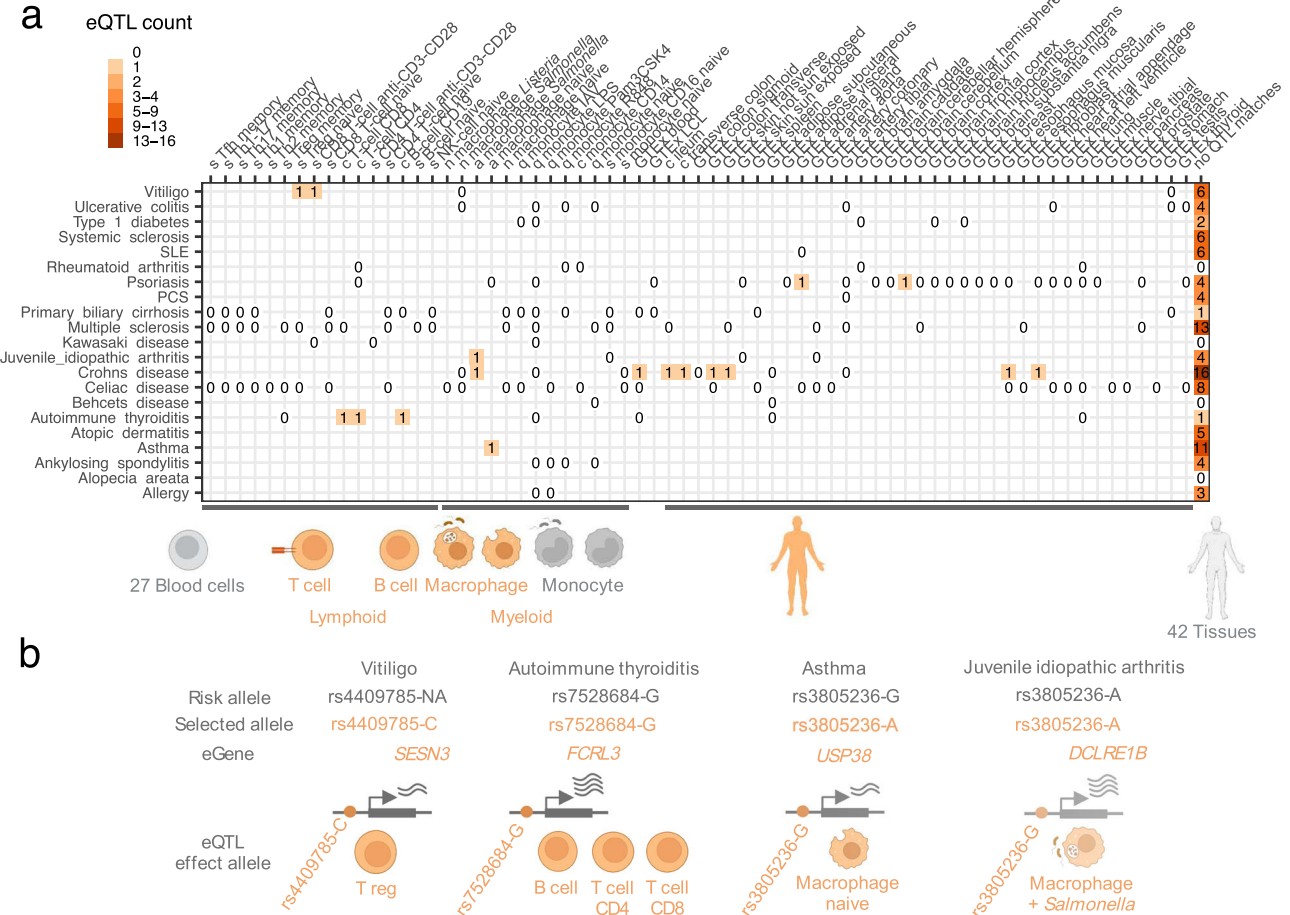

**Fig. 4 | Inflammatory diseases and candidate SNPs under selection. a** Heatmap shows the number of selection targets for each disease (disease eQTLs) among candidate SNPs annotated with eQTL. To provide background, we use '0' to show disease eQTLs that are not under selection. Tissue/cell context for each eQTL is given in columns and highlighted schematically below the heatmap. Full description of tissue/cell context and source studies are given in Supplementary Data 6. Number of candidate SNPs for each disease with no eQTL in currently published studies are summarized in the last column. **b** Selection target eQTLs in blood cells and affected genes. Raw data for all the figure panels are provided in Supplementary Data 6 and Supplementary Data 7. Created with BioRender.com.

region under balancing selection) is the major histocompatibility complex (MHC) locus[38]. This extended locus contains major genetic risk factors (HLA alleles) for many autoimmune conditions[39]. We, therefore, asked if any of the 9102 candidate SNPs outside the MHC locus falls within balancing haplotypes in the human genome. We screened haplotypes with strong evidence (top Beta scores) for balancing selection in European populations[40] and found 565 candidate SNPs (in strong LD with eight index SNPs) that fell within the balancing haplotypes (Supplementary Data 5). These SNPs were missed by the logLR approach, which is not designed to detect balancing selection. Thus, combined with the 153 loci under positive selection, eight loci under balancing selection suggest that around ~30% of the risk loci (161 out of 535) for inflammatory conditions outside MHC contain variants that were important for human adaptation.

**Predicted function for candidate SNPs and selection targets**
Most candidate SNPs for the 21 inflammatory diseases lie outside coding genes[35] and hence non-coding. Here we show that selection targets among these candidate SNPs belong to the same non-coding VEP categories as the rest of the candidate SNPs (Supplementary Fig. 7a). While selection targets were depleted in intergenic regions, we observed no enrichment in coding region VEP categories (Supplementary Fig. 7a). Similarly, no stronger conservation (Supplementary Fig. 7b) or higher deleteriousness (Supplementary Fig. 7c) compared to other candidate SNPs (Supplementary Data 2). Selection targets in

inflammatory disease risk loci are, therefore, more likely to affect gene expression than amino acid sequence. To learn about the affected genes and the cell/tissue context, we compared our 9102 candidate SNPs with cis-acting quantitative trait loci affecting expression (from here onwards, expression quantitative trait loci, eQTLs). Altogether, we screened eQTLs characterized in 50 human body tissues/cells[41,42] and 35 immune cell types and states (naïve, activated, and memory cells) (Supplementary Data 6)[42–46]. By focusing only on statistically significant eQTLs within each tissue and cell type (at a 10% false discovery rate, FDR), we found 919 eQTL-tissue pairs for our candidate SNPs (Supplementary Data 7). We then restricted our search to a narrow set of likely candidate SNPs (PICS > 0.1) and summarized eQTL matches by associated disease and tissue/cell type, where they affect gene expression (Supplementary Fig 8a, Supplementary Data 7). It is notable that these eQTLs belonged to a diverse set of body tissues with no enrichment in immune cells (fisher exact test $p = 0.104$). Nevertheless, when we looked at the affected genes and their biological function, most of them were enriched in immune response pathways, such as, for example, the Th17 axis, which would be expected for autoimmune diseases (Supplementary Fig. 8b). It should be emphasized, however, that most of the candidate variants tested (PICS > 0.1) cannot be explained by the eQTLs in the currently available tissues and cell types (last column on the heatmap, Supplementary Fig. 8a).

Next, for the available top candidate variants with eQTLs, we highlighted selection targets (logLR ≥ 1.59) and reported 17 matches

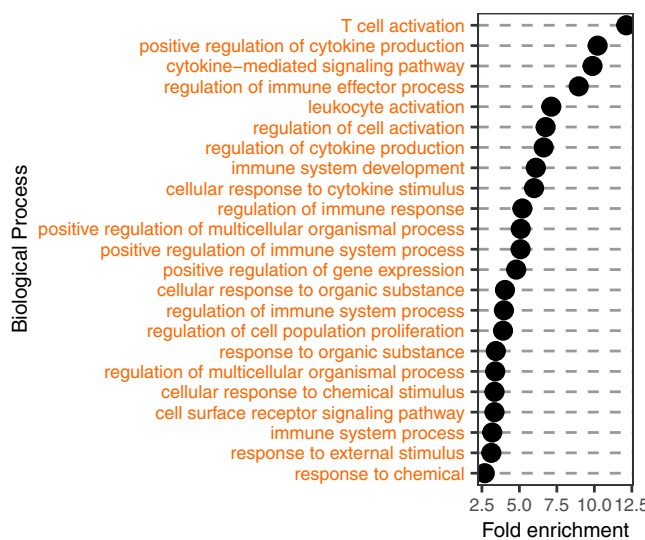

**Fig. 5 | Biological pathways enriched for genes affected by disease eQTLs under selection.** Enrichment of eQTL target genes in GO Biological Process annotation dataset (10.5281/zenodo.639996 Released 2022-03-22) is based on PANTHER Overrepresentation Test[77]. Source data are provided as a Source Data file.

(Fig. 4a, Supplementary Data 6, Supplementary Data 7). These eQTL selection targets were enriched in immune response pathways, such as T cell activation and positive regulation of cytokine production (Fig. 5). We highlighted several of these selection targets with eQTL data from blood cells (Fig. 4b) since they would be easier to experiment using relatively accessible blood samples. For example, the rs7528684-G risk allele for Autoimmune thyroiditis, according to our eQTL data (Supplementary Data 7), increases the expression of the *FCRL3* gene in several immune cells (Fig. 4b), which is also supported by prior evidence. Namely, this rs7528684-G allele was shown to increase *FCRL3* transcription by increasing the affinity of gene promoter to nuclear factor-kB (NF-kB)[47]. Higher expression of the FCRL3 receptor mediated by this risk allele was implicated in many other autoimmune conditions[47], and a recent study on T regulatory cells clarified this protein's role in peripheral tolerance. This transmembrane protein was shown to sense secretory IgA (SIgA), and this mechanism is important to sense microbes at mucosal barriers[48]. Increased FCRL3 production and interaction with secretory IgA switch T regulatory cells towards an inflammatory program, and these cells can no longer suppress CD4 + and CD8 + T cell proliferation, resulting in inflammation. According to our findings, the rs7528684-G allele, associated with this inflammatory phenotype, was under positive selection. Hence, one can hypothesize that this phenotype was beneficial, presumably, due to increased sensitivity to IgA-coated microbes at mucosal barriers of the respiratory and gastrointestinal tracts, the two major routes of infection for deadly pathogens[49]. The flip coin side is, however, the higher propensity to mount inflammatory response when T regulatory cells may encounter IgA-coated commensals at mucosal surfaces. Indeed, IgA-coated commensals are a common feature of dysbiotic gut microbiota in many autoimmune conditions[50].

## Discussion

Pinpointing causal SNPs for complex diseases is challenging, and one underutilized principle is looking for SNPs under natural selection. The rationale is that such variants must strongly affect the trait to be picked by natural selection. Our work was motivated by this idea, and we addressed several issues that hindered its application. We used a novel methodology to show that approximately 28% (153 out of 535) of the risk loci for various inflammatory conditions contain SNPs that were important for adaptation. This fraction is substantially larger than

reported before (21/416 ~ 5%) for 10 inflammatory conditions[13]. We attribute this to a better power of detecting weaker signals of selection with the Relate/CLUES approach. Indeed, when we reproduced the iHS selection test on Estonian dataset, focusing on the same 10 inflammatory conditions, we found that only ~6% of the risk loci had candidate SNPs under selection (29 out of 456 risk loci available in our dataset) (Supplementary Data 2). We also show that while some selection signals are shared across European populations, a sizable portion detected in Estonian Biobank samples was not observed in other tested populations. This finding may indicate region-specific selection signals, and more populations need to be tested. Individual selection targets, therefore, may not be transferable to other populations, even within Europe. Thus, while the principle of prioritizing selection targets is universal, selection targets themselves need to be established for each population separately. Population specificity of positive selection also underscores possible regional differences in environmental exposure and its impact on disease prevalence.

We next asked if we could fine-map selection target mutations in these sweep regions. We showed that fine-mapping is possible by demonstrating that most candidate SNPs within disease risk loci corresponded to weak and moderate sweep scenarios (selection coefficients). Only a few SNPs demonstrated signals of strong selection (Fig. 1a). Weak and moderate selective sweeps leave more nucleotide diversity flanking the selection target mutations. This is in contrast to strong sweep signals that leave little information for fine-mapping and that were reported by earlier studies[12,13]. The inferred weaker selection signals for candidate SNPs also explain why previous studies found fewer sweeps among risk loci[12,13]. For example, we applied the iHS score used in previous studies and confirmed that it recovers only 4% of the risk loci under selection in our Estonian sequence data. Thus, the higher gain is likely explained by better power but not by a higher number of selection signals in the Estonian population. We also note that our neutrality rejection approach helped us detect more selection signals at the cost of incurring small (5%) amounts of false rejections that can be explained by target population demography (Supplementary Fig. 2b). This is a small cost since the goal is to have more promising risk loci with SNPs for experimenting in the future.

The selective sweep scenarios suggested by our analyses motivated us to attempt to fine-map the sweep-driving SNP among the candidate SNPs. By inspecting local logLR variation for candidate SNPs, we show that in ~39% of the risk loci with selection signal (60/153), candidate causal SNPs matched with the likely selection target SNPs. Here, we note that, like in any fine-mapping effort, we prioritize, on average, more than one tightly linked SNP to be a selection target, which is inevitable, and this varies by linkage disequilibrium in the focal loci.

Interestingly, in ~19% (29/153) of the risk loci with selection signal, most likely candidate SNPs were hitchhikers, where either risk allele or protective allele was linked with the selected mutation. Thus, based on extensive empirical data, our study supports the idea that hitchhiking contributes to the high frequency of inflammatory risk variants[51].

In contrast to hitchhiking, the natural selection scenario that we inferred for candidate causal SNPs in 60 risk loci can help address some of the challenges in modern medical genomics. Namely, help to prioritize risk loci and SNPs for functional analyses and find the physiological context[52]. Indeed, most causal variants for chronic inflammatory diseases hide in the non-coding DNA, hindering their fine-mapping within risk loci[35,53], especially when the relevant physiological context is unknown. As briefly mentioned before, the idea is that natural selection picked adaptive variants if they had a strong effect on the molecular traits serving better survival. In our case, we expect these molecular traits to be involved primarily in immune response or in immunometabolism, which ensures immunity at the organismal level[54]. Hence, if adaptive mutations contribute to disease pathogenesis, the strong effect on the underlying molecular trait must be easier to detect and study experimentally. Therefore, when considering

multiple candidates across risk loci or within risk loci, variants with adaptive history can be promising targets to prioritize for functional study. Our work, thus, represents an important milestone in the overall fine-mapping effort in the field.

More importantly, the adaptive model suggests that pathogens or structurally similar symbionts may represent a relevant trigger to elicit the regulatory function of candidate causal SNPs for inflammatory conditions. The regulatory function of such SNPs in immune cells can often go undetected and stay "silent"[55] unless immune cells are triggered by the stimulus which is specific to the disease[56]. In this work, we attempted to link candidate SNPs for inflammatory conditions with a set of predicted regulatory SNPs, the so-called eQTLs. Despite testing eQTLs in a large number of tissues (50 body tissues) and immune cells (35 immune cell types and states), we found only a few matches with 1902 candidate SNPs for the 21 inflammatory conditions (Supplementary Fig. 8a). While alternative explanations are proposed[57], we attribute this to the lack of experiments with disease-specific tissues and triggers among published eQTL datasets (Supplementary Data 6). Indeed, it has been highlighted many times that external stimuli and stimulus-specific regulatory responses are important for discovering immune-related genetic variants in inflammatory conditions[58,59]. Our findings inspired by the adaptive hypothesis offer promising candidates to test the utility of the microbial exposure context in studying causal variant function in autoimmunity.

We conclude our study by highlighting an example of a risk SNP (rs7528684-G) that regulates the expression of a cell surface receptor FCRL3. This receptor is important to sense invading microbes at mucosal barriers, such as the gastrointestinal and respiratory tract. While this sensing mechanism is important to clear pathogens at mucosal barriers, the inflammatory response against mucosal symbionts can be detrimental in the context of autoimmunity. Indeed, when this receptor binds secretory IgA on T regulatory cells, this may, under certain conditions, result in uncontrolled effector T cell response and break immune tolerance[48]. We show that the rs7528684-G allele that increases the production of this surface receptor FCRL3 was under positive selection. Hence, this phenotype associated with pathogen detection was likely beneficial. However, the same allele is associated with many autoimmune conditions and has an adverse effect, albeit at the older age[47]. This phenotype is a promising example of antagonistic pleiotropy that deserves further exploration. We report a number of risk SNPs under positive selection that are associated with late-onset inflammatory diseases (manifesting after the peak of reproductive years) (Supplementary Data 2). Some of them may be explained by antagonistic pleiotropy, provided that these late-onset diseases are associated with improved reproduction or survival during or before the reproductive period[7].

Microbial exposure has long been used to trigger an autoimmune response in animal disease models[23]. There are numerous examples where pathogens (viruses, bacteria, and parasites) are known to trigger human autoimmune disease and exacerbate it further[22]. For example, tonsillar infection with *Streptococcus pyogenes* has been known for decades to trigger and exacerbate psoriasis skin lesions[60–62]. Even in celiac disease, the aberrant response to dietary gluten is initiated by a viral infection[63]. Despite the accumulated evidence, infectious triggers of autoimmune diseases, such as, for example, *Herpesvirus*, are known to be common in a healthy population and do not lead to autoimmunity. Therefore, the major knowledge gap is whether pathogens interact differently with individuals having risk alleles to autoimmune diseases, and that results in autoimmunity. Our study offers an actionable list of 60 candidate risk loci to ask this question for various autoimmune diseases using experiments that involve known microbial triggers[22]. Such experiments can shed light on molecular pathways that microbial exposure initiates in carriers of risk SNPs and illuminate early events in pathogenesis. The latter is a poorly understood aspect of autoimmunity with promises for treatment.

## Methods

### Genetic risk loci associated with autoimmune diseases and LD blocks

In this study, we started with 593 unique genomic risk loci associated with 21 autoimmune disorders and reported selection analyses for 535 loci with MAF > 0.05, for which the previous study prioritized 4950 (4331 unique) potentially causal SNPs ("candidate SNPs"), each annotated with a PICS (Probabilistic Identification of Causal SNPs) score. PICS score reflects the probability of the candidate SNP to be causal given the haplotype structure and observed pattern of association at the locus[35]. As we are interested in evidence for positive selection not only at the potential candidate SNPs but also at linked loci, in addition to the 4331 unique candidate SNPs from ref. [35], we selected 6156 SNPs in high LD (r2 ≥ 0.8) with at least one candidate SNP among the Estonian Biobank samples with whole-genome sequences (see below for details on the dataset used). As a result, we started our analysis with a list of 10487 SNPs (Supplementary Data 1) grouped around 593 unique risk loci (index SNPs) but reported final results for 535 loci with MAF > 0.05. Since risk loci for different diseases can be in linkage disequilibrium and, hence, clustered, we defined LD blocks to track selection signals that can affect such clusters. For that, SNPs were grouped into LD-blocks so that any 2 SNPs belonging to different LD-blocks had r² < 0.6 (Supplementary Fig. 1). For individual risk loci and LD blocks, when risk loci were clustered into LD blocks, we computed haplotypes and their frequencies using the Haplostrips tool version 1.2.1[64]. Disease-specific risk alleles were retrieved either from ref. [35] or from IEU GWAS database using ieugwasr R package version 0.1.5[65]. Handling VCF files with sequence data, filtering, subsetting, as well as LD calculations was done using bcftools version 1.9[66] and vcftools version 0.1.14[67].

### Evidence for natural selection based on local trees

A typical genomic risk locus for a complex disease usually spans several tens of kilobases and contains multiple potentially causal SNPs. Such genomic regions contain smaller chunks bounded by recombination that occurred during the evolutionary history of the risk loci. Each such chunk has a genealogy that correlates with nearby subregions (due to LD). Such genealogies or local trees reflect the full history of the genomic subregion and candidate SNPs in it (Supplementary Fig. 1). If the candidate SNP had a selective advantage in the past, one would expect the selective sweep to distort the local tree and deviate from neutrality expectations in terms of its topology and branch lengths. The extent of distortion can be translated into the likelihood of natural selection and the competing neutrality scenarios. This is the most accurate way to test for evidence of selection. Both elements of this powerful concept are implemented in the recently proposed Relate approach[31] to infer local trees and the CLUES algorithm to compute the log-likelihood of competing models[32]. We used this approach to test evidence for natural selection for candidate SNPs within the 535 risk loci that are clustered into 464 linkage disequilibrium blocks.

Specifically, we first inferred local trees for subregions within each risk loci that cumulatively contained 10487 variants (4331 previously published candidate SNPs with PICS and additional 6156 SNPs in high LD in our sequences). Out of the total 10487 SNPs considered, 9102 passed Relate QC requirements, derived allele frequency (DAF) > 5%, and mapped on the Relate-inferred local trees such that each tree branch contained one SNP (4838) (Supplementary Data 2). Next, for each inferred tree, we estimated the log-likelihood ratio for the positive selection scenario versus the neutral scenario using the CLUES approach. The overall Relate and CLUES-based analyses consist of several key steps briefly outlined below.

To test candidate SNPs for signatures of positive selection, we start by building local genealogies for whole-genome sequences from Estonian Biobank using Relate[31]; We then extract subtrees corresponding to samples of interest. Next, based on the extracted subtrees, we estimate

the coalescence rate through time to trace the trajectory of the effective population size for the Estonian population. This information is needed for CLUES analysis and to mimic the demographic history of the Estonian population when we simulate sequences and explore the logLR distribution expected under neutral demography. Finally, we run the CLUES inference for the SNPs of interest using the extracted subtrees and coalescence rate through time. Below, we describe each step in the Relate and CLUES pipeline in more detail.

## Overview of our approach to detecting positive selection

We chose CLUES[32] for detecting positive selection since it has several desired properties: (1) it can estimate the strength of selection in the form of the selection coefficient (s) and evaluate evidence for selection at the focal SNP in the form of a log-likelihood ratio (logLR) for two competing models, one with the inferred value of s and the other for neutral model (s = 0); (2) it allows selection target fine-mapping by comparing logLR values between partially linked SNPs; (3) it allows the user to specify the time period to focus on and (4) it is suitable for detecting selection on standing variation. This is possible because CLUES estimates the derived allele frequency trajectory over time for a tested SNP using the local tree's topology and branch length as input. A local tree represents evolutionary relationships (the order and time of coalescence) between DNA sequences at a given genomic location. As we move along the chromosome, the topologies and branch length of the local trees change as we cross recombination points; however, overall, neighboring trees are highly correlated in their structure. The properties of a local tree are disturbed by positive selection in a predictable manner, resulting in an increased coalescence rate (and hence shorter length) of branches carrying the selected allele within the time interval when the selection was acting. The local trees and the genome-wide average estimate of coalescence rate through the time required by CLUES as input can be obtained by running Relate - a local tree inference tool that relies on whole-genome sequences[31].

Thus, our inference consisted of the following steps:

1. Preparing the whole-genome sequences data set and applying Relate[31] to it to build local trees.
2. Extracting subtrees corresponding to a subset of samples to (a) filter samples for CLUES and (b) use a smaller subset for estimating coalescence rate over time which becomes computationally costly with too many samples.
3. Running the coalescence rate/branch length estimation procedure on a small subset of the samples but on the whole genome to obtain the neutral expectation of coalescence rate over time.
4. Extracting a local tree corresponding to a given SNP of interest and resampling its branch length to take into account the uncertainty in the branch length estimate. This results in a sample of local trees with the same topology but different branch lengths. As each new tree in the sample is obtained by re-estimating the branch length of the previous tree, the earlier trees in the sample are likely to have less accurate branch length, and hence it makes sense to remove those as burn-in. Also, neighboring trees in the sample are likely to have similar branch length estimates, and hence it is desired to "thin" the sample of trees to avoid redundancy by keeping every ith tree.
5. Running CLUES on the sample of local trees for a given SNP obtained from the previous step.

Steps 1–3 were run once for the whole genome, while steps 4 and 5 were run independently for each SNP of interest. As described above, we started with 10,487 variants, of which 9372 passed the strict callability mask from 1000 Genomes Project[68] and were mapped to Relate-inferred local trees. We then retained 9102 SNPs with minor allele frequency (MAF) > 0.05 for CLUES selection tests. As several SNPs can map to the same branch of the same local tree (reflecting perfect linkage), thus providing the same information, we analyzed one SNP

per branch (4838 in total) and then assigned the same value to all other SNPs residing on the same branch.

## Tree building

The local genealogical trees were built by applying Relate version v1.1.4 to 2420 phased whole-genome sequences of the Estonian Biobank participants described in Kals et al.[33] and Pankratov et al.[34]. We kept only SNP positions with high variant calling certainty in our sequence data using the strict callability mask (GRCh37) from the 1000 Genomes project (The 1000 Genomes Project Consortium). SNP alleles were polarized into ancestral and derived based on the GRCh37_e71 homo sapiens ancestral genome, and SNPs with unknown ancestral state were removed. To carry out the tree-building procedure, we used the GRCh37 recombination map (The 1000 Genomes Project Consortium), the mutation rate of $1.25 \times 10^{-8}$, and the effective population size of 30000 haploids.

## Extracting subtrees

Since CLUES expects a panmictic population, we removed close relatives, ancestry outliers, and clusters of individuals with excessive IBD sharing. For this, we started with the dataset of 2305 samples and removed extreme PCA outliers and close relatives up to 3rd degree. Thus, we removed (a) outliers in PCA within the Estonian dataset or when projecting Estonian samples on the PC space defined by samples from various European populations; in both cases, we removed samples falling out of the 2.5–97.5 percentile range for the first and second Principal Components; (b) top and bottom 2.5% of the samples based on singleton counts per genome and (c) samples that had pairwise total IBD sharing of 166.2 cM or more with more than one another sample in the dataset. After filtering, we randomly subsampled 1800 individuals for CLUES. We further downsampled these 1800 to 100 individuals for coalescence rate through time estimation. Therefore, the subtrees for the subset of 100 and 1800 samples were extracted separately using the SubTreesForSubpopulation mode in the RelateExtract program.

## Estimating coalescence rate through time

To estimate the coalescence rate through time, we applied Relate's EstimatePopulationSize.sh module to the random set of 100 samples with the following parameters: mutation rate $1.25 \times 10^{-8}$, generation time 28 years, number of iterations 5, tree dropping threshold 0.5, and time bins defined as 10x years ago where x changes from 2 to 7 with an increment of 0.1.

## CLUES analyses

Next, we used Relate's local trees corresponding to the 1800 individuals as input for CLUES selection analysis. For a target SNP, we extracted the corresponding local tree and resampled its branch length 200 times. We removed the first 100 sampled trees as burn-in and pruned the remaining 100 trees by keeping every 5th tree. The resulting 20 trees were then used for CLUES. We focused on a time period between 0 and 500 generations ago. As CLUES relies on sampling branch length in an MCMC-like manner, it does not return the same result when run on the same SNP twice, and the degree of discordance depends on the uncertainty in branch length, which is tree-specific. To account for that, we ran CLUES twice for each SNP (starting from branch length sampling), and if the logLR values differed by more than two units (indicating higher uncertainty), we ran it for the third time. Then we assigned the median values of logLR and s (out of 2 or 3 observations) to each SNP.

## CLUES selection test adjusted for genetic drift in population history

To minimize spurious selection signals due to genetic drift, we estimated logLR expected for the Estonian population history using

simulations. Namely, we simulated Estonian demography, computed logLR from it and defined our neutrality rejection criteria using the 95% percentile of the simulation-based logLR distribution (logLR ≥ 1.59) (Supplementary Fig. 2b). For this, we first inferred the effective population size (Ne) trajectory for the Estonian population using Relate (Supplementary Data 8 and Supplementary Fig. 2a) and used parameters of this Ne trajectory to simulate 3600 sequences (1800 diploid individuals) that are equivalent to chromosome 1 (GRCh37) in length using msprime version 0.7.4[69]. Recombination rates were adapted from the GRCh37 recombination map of chromosome 1 (The 1000 Genomes Project Consortium), and the mutation rate was taken as $1.25 \times 10^{-8}$ per nucleotide per generation. To simulate demographic events at generations 0 to 1000, we used the discrete-time Wright-Fisher model (model = "dtwf") and then switched to the Hudson model (model = "hudson") after generation 1000. The simulated sequences were then used to perform tree-building (for 1800 diploids) and Ne trajectory estimation (for a random subset of 100 diploids) as described for the real data. The resulting local tree estimates using Relate were then used to compute logLR using CLUES. To select SNPs for the CLUES analyses, we first removed SNPs that matched at least one of the following criteria: (a) not passing the strict callability mask, (b) not mapping to the inferred trees, and (c) had a minor derived allele count less than four in the sample (1800 diploids), which resulted in 765217 SNPs, altogether. We further pruned our SNP set by retaining every 76th SNP and used the resulting 10,068 SNPs for our CLUES analyses performed as described for the real data. The resulting logLR distribution from simulated data with 95%, 99%, and 99.9% percentiles are shown in Supplementary Fig. 2b.

To compare evidence for selection at candidate SNPs in other European populations, we used published sequence data for British (GBR), Finns (FIN), and Italians (TSI) from the 1000 Genomes Project[68]. LogLR computations were performed using the same Relate/CLUES steps, described above.

### Positive selection test based on iHS score

We scanned traces of positive selection on samples of Estonian Biobank, using the integrated haplotype score (iHS), as implemented in the Selscan program version 1.2.1a[70]. The iHS statistic was computed on SNPs with minor allele frequency (MAF) > 0.05. To allow comparison with earlier studies, we applied the same cut-off threshold as in ref. 13, the top 5% iHS hits in our dataset ($p < 0.05$, |iHS| > 1.9).

### SNP annotations

To obtain functional consequences of candidate SNPs at the transcript, gene, and protein levels, we used Ensembl Variant Effect Predictor (VEP)[71] release 106. We also predicted the deleteriousness of SNPs by retrieving phred-scaled CADD scores release 1.6[72]. Finally, evolutionary conservation at SNP position was estimated using the PhyloP score[73]. PhyloP score shows the substitution rate for each genomic position based on a genomic alignment of 100 vertebrate species. A higher than expected (under neutral drift) substitution rate means acceleration (negative PhyloP score), and a lower rate suggests conservation (positive PhyloP score).

### Inferring eQTLs, target genes and tissues using eQTL Catalogue

We searched our 9102 candidate SNPs in the eQTL Catalogue[74], which stores uniformly standardized eQTL summary data from published sources. Expression QTLs were searched in 50 human body tissues/cells[41,42] and 35 immune cell types and states (naïve, activated, and memory cells)[42–46] (Supplementary Fig. 8). For each tissue, we filtered significant eQTLs at 10% false discovery rate (FDR) using qvalue R package version 2.28.0[75]. eQTLs at 10% FDR were then matched with candidate SNPs. Enrichment in hematopoietic cell types was tested using fisher exact test in the stats R package[76]. Enrichment analysis of

biological pathways for eQTL affected genes was performed using the PANTHER Overrepresentation Test[77] on http://pantherdb.org/tools/compareToRefList.jsp. Specifically, we tested eQTL target genes against the full Homo sapiens reference genes using the GO Biological Process annotation dataset (https://doi.org/10.5281/zenodo.6399963 Released 2022-03-22).

### Basic statistical tests and graphs

Spearman correlation (rho) was computed using the stats package function cor.test in R version 4.2.0[76]. Graphs were produced using the ggplot2 package version 3.3.6[78] in R.

### Reporting summary

Further information on research design is available in the Nature Portfolio Reporting Summary linked to this article.

### Data availability

The Estonian Biobank sequencing data analyzed in this study are available upon request. The application procedure to access the data can be found under the following link: https://genomics.ut.ee/en/content/estonian-biobank Full annotation data for the 153 risk loci with logLR, PICS scores, risk haplotype, Ensembl genes, and LD blocks are available in the Figshare repository with the identifier https://doi.org/10.6084/m9.figshare.16691638.v1. Source data are provided with this paper. The GO Biological Process annotation dataset used in this study is available in https://zenodo.org with the identifier https://doi.org/10.5281/zenodo.6399963. The 1000 Genomes Project strict mask files are publicly accessible through http://ftp.1000genomes.ebi.ac.uk/vol1/ftp/release/20130502/supporting/accessible_genome_masks/StrictMask/. The Ancestral Genome sequences for Homo sapiens (GRCh37) are accessible through http://ftp.1000genomes.ebi.ac.uk/vol1/ftp/phase1/analysis_results/supporting/ancestral_alignments/. The 1000 Genomes Project Recombination map is publicly available through http://ftp.1000genomes.ebi.ac.uk/vol1/ftp/technical/working/20110106_recombination_hotspots/. The CADD score version 1.6 is publicly available through https://cadd.gs.washington.edu/download. The PhyloP measure of evolutionary conservation is available through http://hgdownload.soe.ucsc.edu/goldenPath/hg19/phyloP100way/. The OpenGWAS database of GWAS summary datasets is accessible through https://gwas.mrcieu.ac.uk/. The eQTL Catalogue database of uniformly processed eQTL datasets is accessible through https://www.ebi.ac.uk/eqtl/. The Ensembl VEP release 106 is accessible through https://www.ensembl.org/info/docs/tools/vep/index.html. Source data are provided with this paper.

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

## Acknowledgements

This research was supported by the European Union through the European Regional Development Fund (Project No. 2014-2020.4.01.16-0125, Project No. 2014-2020.4.01.16-0030, Project No. 2014-2020.4.01.15-0012, MOBEC008), Estonian Research Council (grants PRG243), Russian Foundation for Basic Research (grant 18-04-00972\18). V.P. was supported by the European Union through Horizon 2020 research and innovation program under grant no 810645. B.Y. was supported by the Government of the Russian Federation through the ITMO Fellowship and Professorship Program. We thank Leo Speidel and Aaron Stern for their guidance and help with running Relate and CLUES analyses. Data analyses for this study were carried out in the High-Performance Computing Center of the University of Tartu. Figure 4 and Supplementary Fig. 8 were created with BioRender.com.

## Author contributions

Conceived and designed the experiments: B.Y., M.Y. Analyzed the data: B.Y., V.P., M.Y., and S.R., Interpreted and Discussed results: V.P., M.Y., B.Y. Implemented simulation study: V.P., M.Z. and S.R. Contributed reagents/materials/analysis tools: M.Z., S.R., V.P. Wrote the paper: B.Y., M.Y., and V.P. Estonian Biobank Research Team collected and provided sequence data on Estonian Biobank donors.

## Competing interests

The authors declare no competing interests.

## Ethics approval

This study used whole genome sequence data on Estonian Biobank participants generated in ref. 33. The original study was approved by the Research Ethics Committee of the University of Tartu (application number 234/T-12). All Estonian Biobank participants have signed a broad informed consent which allows research in the fields of genetic epidemiology, disease risk factors and population history.

## Additional information

## Estonian Biobank Research Team

**Andres Metspalu[4], Mari Nelis[4], Lili Milani[4], Reedik Mägi[4] & Tõnu Esko[4]**

[4]University of Tartu, Institute of Genomics, Tartu 51010, Estonia.

