## [Peer Review File · Nature Communications]

Prioritizing Autoimmunity Risk Variants for Functional Analyses by Fine-Mapping Mutations Under Natural SelectionREVIEWER COMMENTS

Reviewer #1 (Remarks to the Author):

In this manuscript, the authors prioritize genetic variants under natural selection in loci previously reported as associated with inflammatory diseases, under the premise that such prioritized variants might become targets of future functional experiments. Unlike previous reports of modest (~5% - 10%) evidence for natural selection in variants within inflammatory disease loci, these analyses show that that ~37% of the risk loci for various inflammatory conditions contain at least one SNP that demonstrates evidence for natural selection. Among these loci, ~28% of the candidate functional SNP colocalized with the SNP with evidence for selection, and a large fraction (60%) of candidate variants were either hitchhikers (~27%) or linked with the SNP with evidence for selection (~20%).

This work is significant to all who work on population genetics of autoimmune or inflammatory traits, those who seek to understand whether and how population genetics phenomena like natural selection might underlie the frequencies of alleles that predispose to these disorders, and those interested in fine-mapping autoimmune or inflammatory disease loci. This work expands on previous work by applying a recently developed approach to test for evidence for natural selection for candidate SNPs within previously reported 593 inflammatory disease loci in the Estonian population. The results unveil that a fraction larger than that previously reported for these loci show evidence for natural selection, and demonstrate that alleles associated with inflammatory traits might have risen to higher frequencies in the population due to genetic hitchhiking. In addition, the SNPs with evidence for selection identified in this study are promising candidates for functional studies.

Although overall strong, several claims or conclusions are overstated and need clarification. For example, what specifically are the authors referring to when stating that “we analyzed the largest collection of risk loci to date” (line 225)? The list of autoimmune GWAS SNPs published by Fahr and colleagues in 2015 might be the largest published, but is also outdated (it used SNPs downloaded from the NHGRI-EBI GWAS Catalog in 2013).

More importantly, this study focuses solely on variants with evidence for natural selection in the Estonian population, without properly acknowledging that the reported SNPs are only adaptive in this population, without acknowledging the varying population differences in autoimmune disease prevalence, and without discussing the population-specific effects of natural selection and of autoimmune disease-associated genes. Although the results are presented as “universal”, the effects of adaptive and causal variants in different populations need to be discussed.

Although the analytic details of the methodology and algorithms used are outside my area of expertise, the quality of the data and validity of the approach seems robust. Specifically, the Relate approach by Speidel et al (2021) to construct coalescent trees, and the CLUES algorithm by Stern et al (2019) to infer selection coefficient and historical allele frequency, are sound approaches to test for evidence for natural selection for candidate SNPs.

The sequencing data from the Estonian Biobank is available upon request, and several analytical details seem to have been previously reported (e.g. the exclusion of “close relatives, ancestry outliers, and clusters of individuals with excessive IBD sharing” from Estonian Biobank participants). However, the code used by the authors was not made available. The details provided in the methods are thus partially sufficient to allow reproduction of the analyses described.

To increase the clarity, provide context for the results, and help with result interpretation, I have the following suggestions:

- 1) Please acknowledge and discuss population or geographic effects on the results. Since the results report evidence for selection only in the Estonian population, please comment on the generalizability of the results to non-Estonian populations with varying demographic histories and autoimmune

disease prevalence.

2) I concur with the authors that the approach used has better power of detecting weaker signals of selection compared to other approaches. However, it's not clear how the approach alone can justify the increase of autoimmune disease loci harboring genetic variants under natural selection from previously reported ~5-10% to ~37% (while Raj et al (2015) reported that inflammatory diseases have ~5% of SNPs targeted by positive selection, Ramos (2017) reported that autoimmune rheumatic diseases have ~10% of SNPs targeted by positive selection). Since disease causal effect sizes appear to be most population-specific for immune genes and regions under selection (Shi et al, Nat Comm 2021), could the results be more Estonian-specific? Also, can the authors elaborate on the biologic meaning of this result? For example, how might this higher frequency of inflammatory loci harboring variants under natural selection explain the varying prevalence of these conditions in different populations? How can this be interpreted in light of the higher inflammatory responses observed in individuals of African ancestry (Nedelec et al, 2016)?

3) Please make the code used for the analyses also available.

4) The Introduction is slightly long; could it be summarized?

5) The interchangeability between "mutation" and "variation" or "SNP" is not the most accurate terminology (mutation and variation are not the same). Please define or harmonize.

6) For consistency, please use "Relate" (not both "Relate" and "RELATE"). Also, celiac disease isn't capitalized.

7) In the Abstract (line 30), please replace: "Such experiments will help decipher molecular events triggered by infectious agents...", to "...potentially triggered by infectious agents", as defining the exact causal agent is difficult.

8) Similarly, in lines 52 and 109, please revise the sentences "While the adaptive history of risk alleles for autoimmunity will offer new opportunities to understand their function under microbial invasion", and "the key contribution is that we identify SNPs that are promising candidates for experimental study of the causal SNP function under microbial exposure context", as in addition to pathogens, other exposures such as diet, altitude, radiation levels, or temperature might also affect adaptation.

9) The reference for Marnetto et al, 2021 (line 443) is not in the bibliography ("Note that the overall approach we use here to run CLUES is the same as in Marnetto et al, 2021").

10) In Table 1, it would be helpful to include the gene region.

11) In Figure 2, the font is too small, one can't assess what is shown; the legend is confusing, interchanging between "graph" and "line" - "row" would perhaps help. For example, "The third line from the top shows candidate SNP rsID along with the disease effect allele" actually looks like the second line to me. For more intuitive interpretation, I'd also suggest a gradient between similar colors for the r-squared, instead of jumping from pink ($r^2=1.0$) to white ($r^2=0.75$) to blue ($r^2=0.5$).

12) In Figure 3, it would help to explain in the legend what the numbers on the left arrows and 5 bottom arrows are. Also, I count 208, not 204 loci...

13) Supplementary Tables 1 and 2 are truncated.

14) In Supplementary Figures 3 and 4, the IDs on the "regulatory elements" panel are overlapping.

Reviewer #2 (Remarks to the Author):

The work by Pankratov et al is innovative since they show that selection pressures on autoimmune related genetic variants are more frequent than previously thought. They started with 4331 SNPs that were reported by Farh et al., 2015, as potential causal variants for 593 GWAS hits in 21 autoimmune conditions and they also included 9151 more that are in high LD with the 4331 ones. By using modern algorithms (RELATE, CLUES) they managed to fine map SNPs that are under weak or moderate selection, or they are in LD with such SNPs. They claim that their results can contribute to future functional experiments for significant autoimmunity related variants. The methodology they used is sound and their results are of significance in the field of autoimmune genetics.

Before publication, there are some points that must be improved.

Major issues:

1. The authors could also perform as a second method the more classical test "Tajima D", for results confirmation
2. Can the algorithms used by the authors distinguish between genetic drift and selection?
3. The authors do not discuss at all the genes under study, at least some of them, where positive selection was found. Can they propose the most important of them for future functional studies?
4. A table is missing summarising the most significant results they found, paying attention at non-synonymous SNPs
5. Are the non-synonymous SNPs under weak or moderate selection conserved?

Minor issues:

1. Line 42: It would be fair the authors to complete that antagonistic pleiotropy idea was mostly developed by George Williams (1957). They could also discuss a little more (in Discussion) about the significance of their results for antagonistic pleiotropy
2. The authors repeatedly talk about hitchhiking throughout the text, but they have to explain more clearly that genetic hitchhiking is actually the Linkage Disequilibrium phenomenon

We would like to thank the referees for their time, for carefully reading our manuscript, and for giving constructive comments, which helped to improve the manuscript substantially.

It is important to note that in response to numerous questions and suggestions from both referees, we added new analyses and substantially updated the manuscript. We added figures and tables to present new analyses and support some of the old findings.

Reviewer 1

Reviewer Comment	Response
For example, what specifically are the authors referring to when stating that “we analyzed the largest collection of risk loci to date” (line 225)? The list of autoimmune GWAS SNPs published by Fahr and colleagues in 2015 might be the largest published but is also outdated (it used SNPs downloaded from the NHGRI-EBI GWAS Catalog in 2013).	A more accurate formulation would be "the largest collection of fine-mapped candidate SNPs for the largest collection of inflammatory conditions." We have now updated this sentence. Lines 219-220. We agree that newer versions of the NHGRI-EBI GWAS catalogue offer a larger set of risk loci. However, for our study purposes, we needed uniformly processed candidate SNPs so that there would be SNP by SNP information for predicting the probability of each SNP being causal. The Fahr et al. study offers such a dataset. Namely, it offers SNP-wise PICS scores for the largest collection of risk loci (over 535) across 21 diseases. For comparison, the latest compilation of fine-mapped autoimmune disease loci features only 230 genomic regions for 15 conditions (Caliskan et al. 2021, AJHG).
More importantly, this study focuses solely on variants with evidence for natural selection in the Estonian population, without properly acknowledging that the reported SNPs are only adaptive in this population, without acknowledging the varying population differences in autoimmune disease prevalence, and without discussing the population-specific effects of natural selection and of autoimmune disease-associated genes. Although the results are presented as “universal”, the effects of adaptive and causal variants in different populations need to be	This is an important point, and we have added new analyses to explore population specificity. In the revised manuscript, we evaluated if selection targets in Estonians are shared with other European populations and to what extent. We used both the SNPs under selection detected in Estonians and the full set of candidate disease SNPs not under selection. Both SNP sets were tested using the Relate/CLUES approach on sequence data from other European populations. We used closely related Finns, British, and

discussed.

distantly related Italians. The SNP-by-SNP frequencies and logLR estimates were compared between populations using Spearman's rank correlation analysis (ρ). We first confirmed that candidate SNP frequencies, when taken together, are pretty similar between European populations (Figure 1d, high ρ based on allele frequencies). This is because most of the common alleles drifted little since these populations separated. On the other hand, if most polymorphisms drifted neutrally, logLR values are expected to vary around 0. Indeed, we observed little correlation in logLR for most candidate SNPs (Figure 1e). For example, $\rho=0.19$ between Finns (FIN) and Estonians (EST) when all the SNPs were taken ($\log LR > 0$) (Figure 1e). We then separately considered SNPs under selection in Estonians that potentially underwent shared or convergent selection in other Europeans (orange dots in Figure 1e at $\log LR > 1.59$ and Supplementary Fig.3 at $\log LR > 3$). While correlation increased as we progressively filtered stronger signals at $\log LR > 1.59$ and $\log LR > 3$, the overall concordance remained low (Figure 1e and Supplementary Fig. 3). Thus, a sizable portion of selection signals in Estonians are likely population-specific. Also, selection signal sharing decreases as we consider more distant populations: $\rho=0.3$ for Finns vs and Estonians > $\rho=0.28$ for British vs Estonians > $\rho=0.15$ for Italians vs Estonians (Figure 1e).

We added a new section to describe these results.

Lines 159-175.

We also briefly discussed the population specificity of the discovered selection signals and their implications.

Lines 334-340.

However, we could not find reviews/published resources to connect allele frequency differences, differences in exposure, and

	autoimmunity prevalence across European populations. We felt we have little data to expand on this aspect in this work.
The sequencing data from the Estonian Biobank is available upon request, and several analytical details seem to have been previously reported (e.g. the exclusion of “close relatives, ancestry outliers, and clusters of individuals with excessive IBD sharing” from Estonian Biobank participants). However, the code used by the authors was not made available. The details provided in the methods are thus partially sufficient to allow reproduction of the analyses described.	We have added a Supplementary Materials section Supplementary Information with free software and databases that we used to run all the analyses.
To increase the clarity, provide context for the results, and help with result interpretation, I have the following suggestions: 1) Please acknowledge and discuss population or geographic effects on the results. Since the results report evidence for selection only in the Estonian population, please comment on the generalizability of the results to non-Estonian populations with varying demographic histories and autoimmune disease prevalence.	To provide context on the generalizability of our inferences, we compared natural selection signals detected in Estonians to that in other European populations. SNP-wise comparison showed low correlation (Figure 1e and Supplementary Fig. 3). While part of the discrepancy might be due to variation in logLR estimates, which needs further exploration, it seems that selection targets discovered in Estonians are only partially transferable to other European populations. We acknowledge this limitation in this revised version. Nevertheless, we note that the principle of focusing on selection targets is universal and would be interesting to apply to other populations with Biobank-scale sequence datasets. Lines 159-175. Lines 334-340.
2) I concur with the authors that the approach used has better power of detecting weaker signals of selection compared to other approaches. However, it’s not clear how the approach alone can justify the increase of autoimmune disease loci harboring genetic variants under natural selection from previously reported ~5-10% to ~37% (while Raj et al (2015) reported that inflammatory diseases have ~5% of SNPs targeted by	In this revised version, we evaluated whether Estonians have an unusually high amount of selection signals in autoimmune risk loci compared to other populations. To facilitate comparison, we analyzed candidate SNPs for the 456 risk loci associated with ten inflammatory conditions reported in Raj et al., 2013. We applied the same iHS-score method and significance

positive selection, Ramos (2017) reported that autoimmune rheumatic diseases have ~10% of SNPs targeted by positive selection). Since disease causal effect sizes appear to be most population-specific for immune genes and regions under selection (Shi et al, Nat Comm 2021), could the results be more Estonian-specific? Also, can the authors elaborate on the biologic meaning of this result? For example, how might this higher frequency of inflammatory loci harboring variants under natural selection explain the varying prevalence of these conditions in different populations? How can this be interpreted in light of the higher inflammatory responses observed in individuals of African ancestry (Nedelec et al, 2016)?

criteria as Raj et al. 2013 who reported ~5% risk loci under selection. Specifically, we classified risk loci to have a selection signal if candidate SNPs were among the top 5% iHS scores (2.5% from both tails of the distribution). When all the candidate SNPs (across 456 unique risk loci) were considered, we found 29 risk loci to have SNPs in the top 5% iHS hits in the Estonian dataset, i.e., significant at $p < 0.05$ ($|iHS| > 1.9$). This threshold is almost identical to the one used in Raj et al. 2013 ($|iHS| > 2$). Thus, with the same approach as in Raj et al., we find that Estonians have ~6% of risk loci under selection (29/456~0.06), which is similar to other European populations (Suppl Table 2). We conclude that there is no Estonian-specific strong enrichment in selection signals. Instead, we show that the increase in the number of selection signals is attributable to the higher sensitivity of the CLUES approach for weaker selection signals, as shown in Figure 1a.

Lines 330-334.

Finally, we updated our analyses to reduce spurious selection signals. We constructed LD blocks and visually inspected LD between candidate SNPs with $\log LR > 1.59$ and required these SNPs to show a consistent selection signal (schematically outlined in Figure 2a). In this way, we retained 153 risk loci (instead of 204 as before) with consistent evidence for selection.

Lines 121-126.

Regarding the second part of the question, to the best of our knowledge, Estonians do not have a higher prevalence of autoimmune diseases, at least at the scale reported for intercontinental groups, such as Africans versus Europeans (Brinkworth and Barreiro, 2014). Frankly speaking, we find it problematic to assess the contribution of selection targets among risk loci to the

	disease prevalence. First, selection targets among risk loci comprise only a fraction of the disease's genetic predisposition. Secondly, we do not know how to convincingly tease apart the portion of the genetic predisposition attributable to these selected SNPs and then estimate its' contribution to prevalence, bearing in mind the GxE (genotype and environment) interaction.
3) Please make the code used for the analyses also available.	In this study, no custom software code was developed. Instead, we used freely available software code and databases, which we now listed in a separate Supplementary Methods section of Supplementary information.
4) The Introduction is slightly long; could it be summarized?	We shortened the Introduction (57 lines in the revised version versus 74) by summarizing some of the ideas, removing redundancy, and trimming wordiness. Other sections were also trimmed, where possible.
5) The interchangeability between “mutation” and “variation” or “SNP” is not the most accurate terminology (mutation and variation are not the same). Please define or harmonize.	We spotted some examples of inaccurate use, such as in the Discussion "Pinpointing causal mutations for complex diseases." These were fixed. We tried to harmonize as follows. We retained both variant and SNP as both are commonly used to mean disease-causing variants. We also retained mutation when it refers to adaptive change since it is commonly used in the field of evolutionary biology.
6) For consistency, please use “Relate” (not both “Relate” and “RELATE”). Also, celiac disease isn’t capitalized.	Done
7) In the Abstract (line 30), please replace: “Such experiments will help decipher molecular events triggered by infectious agents...”, to “...potentially triggered by infectious agents”, as defining the exact causal agent is difficult.	Done
8) Similarly, in lines 52 and 109, please revise the sentences “While the adaptive history of risk alleles for autoimmunity will offer new	Both sentences were revised to acknowledge the uncertainty about the selection agent. In addition, we substantially updated and

opportunities to understand their function under microbial invasion”, and “the key contribution is that we identify SNPs that are promising candidates for experimental study of the causal SNP function under microbial exposure context”, as in addition to pathogens, other exposures such as diet, altitude, radiation levels, or temperature might also affect adaptation.	shortened the Introduction. Lines 50-55 and 86-90.
9) The reference for Marnetto et al, 2021 (line 443) is not in the bibliography (“Note that the overall approach we use here to run CLUES is the same as in Marnetto et al, 2021”).	This missing reference was removed since details on the CLUES approach are self-sufficient.
10) In Table 1, it would be helpful to include the gene region.	Genomic regions for genes were included. Given the new analyses and expanded scope, we moved Table 1 to Supplementary Table 4 in the interest of space.
11) In Figure 2, the font is too small, one can't assess what is shown; the legend is confusing, interchanging between “graph” and “line” – “row” would perhaps help. For example, “The third line from the top shows candidate SNP rsID along with the disease effect allele” actually looks like the second line to me. For more intuitive interpretation, I'd also suggest a gradient between similar colors for the r-squared, instead of jumping from pink ($r^2=1.0$) to white ($r^2=0.75$) to blue ($r^2=0.5$).	We updated the entire figure and legend, as well as the LD heatmap, as suggested. In this revised manuscript, Figure 3 replaced the old Figure 2.
12) In Figure 3, it would help to explain in the legend what the numbers on the left arrows and 5 bottom arrows are. Also, I count 208, not 204 loci...	We substantially updated the old Figure 3 (It is now Figure 2b) by adding panels a and b that describe the overall flow of the analyses. We also provide a more detailed legend, as suggested. The number of risk loci under selection was also updated.
13) Supplementary Tables 1 and 2 are truncated.	These supplementary tables were updated and uploaded under the correct file category to avoid truncation. Changes: Supplementary Table 1 > Supplementary Table 2 Supplementary Table 2 > Supplementary

	Table 5
14) In Supplementary Figures 3 and 4, the IDs on the “regulatory elements” panel are overlapping.	Since a large number of regulatory elements can't be accommodated within these graphs, we removed these annotation rows altogether. Figures were renumbered: Supplementary Figures 4 and 5

Reviewer 2

Reviewer Comment	Response
Major issues	
1. The authors could also perform as a second method the more classical test “Tajima D”, for results confirmation	This is an important point. We considered the classical Tajima D and a more commonly used iHS statistics. It turned out that Tajima D is biased when applied to growing populations. We show that Estonians experienced rapid population growth (Supplementary Figure 2a), and this would bias Tajima D towards negative values. We, therefore, chose iHS, which was used by others to discover selection in autoimmune risk loci (Raj et al., 2013). In addition, iHS outperforms Tajima D (Voight et al., 2006) in detecting positive selection. A comparison of iHS with logLR was now added to the main text (Supplementary Figure 6 and Supplementary Table 2). Overall, we find a weak correlation between the two methods. We show that iHS inferences agree with the more powerful logLR test only for the strongest selection signals (Supplementary Figure 6e) but show a low correlation over much of the selection signal space (weak to strong logLR values). We also use these findings to discuss possible reasons for the discrepancy with the published iHS-based inferences. Of course, a deeper understanding of this discrepancy would require a dedicated study with extensive simulations.

	Lines 224-256.
2. Can the algorithms used by the authors distinguish between genetic drift and selection?	Spurious selection signals due to genetic drift in the target population were accounted for using our simulation study. Namely, we first reconstructed the demographic history for the studied population (Estonians) in terms of the N_e (effective population size) trajectory, which showed ancient bottleneck events and recent growth (Supplementary Fig. 2a). This inferred N_e history was used to simulate Estonians' neutral demographic history. We then computed logLR from simulated data and defined the neutrality threshold using the 95% percentile of the simulated neutral distribution (Supplementary Fig. 2b). For Estonians, we found that the 95% percentile at $\log LR = 1.59$ yields approximately 5% false rejections due to neutral demography. In this revised manuscript, we now describe more explicitly this procedure to account for the effect of genetic drift in the studied population. Lines 111-115. Lines 574-597.
3. The authors do not discuss at all the genes under study, at least some of them, where positive selection was found. Can they propose the most important of them for future functional studies?	This is an important problem in the field of complex disease genetics since most disease candidate SNPs, and selection targets among them are non-coding. We tried to address this challenging problem and report affected genes for selected SNPs using two approaches. First, we inferred target genes by looking for eQTLs in 85 tissues (Supplementary Table 6) that were stored in the eQTL Catalogue. Altogether, we tested eQTLs in 50 human body tissues/cells and 35 immune cell types and states (naïve, activated, and memory cells). In Supplementary Table 7, we highlight promising SNPs for functional studies. They can be filtered using 'TRUE' flag in column 'Promising SNPs on A-C LD-blocks ($\log LR \geq 1.59$ and $PICS \geq 0.1$)'.

	Also, in this revised version, we summarised tissues (Figure 4a) and biological pathways for genes affected by selected SNPs (Figure 5). Finally, we highlight examples of selected SNPs that have eQTLs in blood cells since they would be relatively easy to test experimentally (Figure 4b). We also report the nearest genes for candidate SNPs in Supplementary Table 7 that were originally inferred in Farh et al. 2015. A new section was added in Results to describe these findings. Lines 279-321.
4. A table is missing summarising the most significant results they found, paying attention at non-synonymous SNPs	We have now accompanied all our major analyses with supplementary tables. For instance, results from different analyses are integrated into Supplementary Table 2. There, we put together Relate/CLUES selection tests for each SNP, iHS-score for comparison, its conservation, a functional consequence at the transcript, gene, and protein level, as well as other details. More importantly, we report promising candidate SNPs for functional follow-up in column 'Promising SNPs on A-C LD-blocks (logLR \geq 1.59 and PICS \geq 0.1)' that can be selected by filtering 'TRUE' values.
5. Are the non-synonymous SNPs under weak or moderate selection conserved?	We compared phyloP evolutionary conservation metric for coding (synonymous and nonsynonymous) and non-coding SNPs under selection and those not under selection (Supplementary Figure 7b). For example, missense SNPs under selection did not show any difference in conservation compared to nonsynonymous SNPs (Supplementary Figure 7b). Also, the few coding SNPs under selection did not show any marked increase in conservation (positive PhyloP) compared to non-coding SNPs. The limitation here is that we have only a small number of coding SNPs among disease variants. We, therefore, do not consider these findings conclusive. Nevertheless, we add this comparison in the

	main text. Lines 273-279.
Minor issues	
1. Line 42: It would be fair the authors to complete that antagonistic pleiotropy idea was mostly developed by George Williams (1957). They could also discuss a little more (in Discussion) about the significance of their results for antagonistic pleiotropy	Many thanks for this clarification - we have now included a reference to this landmark work. Our new findings based on eQTL analyses (Figure 4 a,b) allowed us to highlight a promising example of potential antagonistic pleiotropy that deserves further exploration. Namely, the biology behind the rs7528684-G risk variant. This SNP increases the risk of many autoimmune conditions by increasing the production of the FCRL3 receptor. An increased number of these surface receptors increases the sensitivity of regulatory T cells to secretory IgA, which coats microbial agents on mucosal barriers. This results in a strong T-cell response at mucosal barriers (lung and gastrointestinal tract) and inflammation. We show that this allele was under positive selection. Hence, this phenotype was likely beneficial to survival, presumably because it conferred higher sensitivity to deadly pathogens at the major sites of infection in humans - lung and gastrointestinal tract. The cost of this adaptation is an undesired response to commensals coated by SIgA. Indeed, SIgA-coated commensals are common in the dysbiotic gut in autoimmune patients. We added these findings to Results and Discussion: Lines 303-321. Lines 397-412.
2. The authors repeatedly talk about hitchhiking throughout the text, but they have to explain more clearly that genetic hitchhiking is actually the Linkage Disequilibrium phenomenon	Indeed, the term hitchhiking is mostly used by population geneticists. We now introduce this term at its first occurrence: Line 43-46 and 84-86.

REVIEWERS' COMMENTS

Reviewer #1 (Remarks to the Author):

I would like to thank the authors for their responsiveness and congratulate them for this important work.

I only have one concern: in line 336, stating that some selection signals are "population specific to the Estonians" might be interpreted by the general public (and any non-human genetics audience) as if these genetic signals were exclusive, unique and limited to those born in Estonia. To avoid the dangerous assumption that individuals can be precisely circumscribed into distinct and homogeneous groups (which is at the basis of racist and eugenic ideologies), using expressions that denote gradients of variation, like "higher frequency", are preferable. I would thus recommend re-wording this sentence trying to convey the relative enrichment of selection signals in Estonians.

On a personal, follow-up note, there are reviews that note a North-South gradient of decreasing prevalence of a few autoimmune diseases in Europe (a quick search yielded Frazzei et al. Front Immunol. 2022;13:899372). Although it would be interesting to understand whether natural selection has contributed to this varying prevalence, it is outside of the scope of this manuscript.

In sum, all of my questions and suggestions were properly addressed by the authors, and I am very pleased with the response.

Reviewer #2 (Remarks to the Author):

The authors have substantially improved their manuscript according the comments of both Reviewers. I am satisfied with the authors' answers and I suggest publication.

Response to referees

Reviewer 1

Reviewer Comment	Response
I would like to thank the authors for their responsiveness and congratulate them for this important work. I only have one concern: in line 336, stating that some selection signals are “population specific to the Estonians” might be interpreted by the general public (and any non-human genetics audience) as if these genetic signals were exclusive, unique and limited to those born in Estonia. To avoid the dangerous assumption that individuals can be precisely circumscribed into distinct and homogeneous groups (which is at the basis of racist and eugenic ideologies), using expressions that denote gradients of variation, like “higher frequency”, are preferable. I would thus recommend re-wording this sentence trying to convey the relative enrichment of selection signals in Estonians. On a personal, follow-up note, there are reviews that note a North-South gradient of decreasing prevalence of a few autoimmune diseases in Europe (a quick search yielded Frazzei et al. Front Immunol. 2022;13:899372). Although it would be interesting to understand whether natural selection has contributed to this varying prevalence, it is outside of the scope of this manuscript.	First of all, we would like to thank the referee for his time and feedback that helped us to improve the quality of the manuscript. As suggested, we rephrased this sentence and the following sentences to avoid ambiguity in the interpretation. Lines 331-336 Thank you for directing us to this work. We agree and expect that possible correlations between disease prevalence and natural selection could be better understood in future as more sequencing in other parts of Europe, the Middle East, and Africa will soon make it possible to get a detailed map of natural selection.

Reviewer 2

Reviewer Comment	Response
The authors have substantially improved their manuscript according the comments of both Reviewers. I am satisfied with the authors' answers and I suggest publication.	We are happy to hear our work is recommended for publication. We would like to thank the referee for his time and efforts in helping us improve the manuscript.